# Port Placement Variations for Robotic Lung Resection: Focusing on Their History, Conventional Look-Up-View and Horizontal Open-Thoracotomy-View Techniques, and More

**DOI:** 10.3390/jpm13020230

**Published:** 2023-01-27

**Authors:** Noriaki Sakakura, Takashi Eguchi

**Affiliations:** 1Department of Thoracic Surgery, Aichi Cancer Center Hospital, Nagoya 464-8681, Japan; 2Division of General Thoracic Surgery, Department of Surgery, Shinshu University School of Medicine, Matsumoto 390-8621, Japan

**Keywords:** approach, history, port placement variation, robotic lung resection

## Abstract

This is a narrative review that summarizes the variations in approaches and port placements used for performing robotic lung resections on the da Vinci Surgical Platforms. Currently, the four-arm, look-up-view method, in which the intrathoracic cranial side is viewed from the caudal side, is considered the mainstream approach worldwide. Several variations were devised from this conventional technique, including the so-called horizontal open-thoracotomy-view techniques in which the intrathoracic craniocaudal axis is aligned with the horizontal direction of the console monitor, and fewer port and incision techniques. In September 2022, 166 reports were surveyed using a PubMed English literature search, and this review finally included 30 reports describing the approaches. We categorized the variations into four-phase groups considering advent histories: (I) early era, three-arm technique with utility incisions; (II) four-arm, total port technique without robotic staplers; (III) four-arm technique using robotic staplers; (IV) maximizing the functional features of the Xi, significant alterations in viewing directions, and reducing ports, including the ultimate uniport technique. To comprehensibly visualize these variations for practical use, we created elaborate illustrations based on the literature. The familiarity of thoracic surgeons with the variations and characteristics allows them to choose the optimal procedure that best suits each patient and their preferences.

## 1. Introduction

This narrative review briefly summarizes and depicts the variations in the main port placements and approaches used for performing robotic lung resections (RLRs) in robot-assisted thoracoscopic surgery (RATS) on the da Vinci Surgical System platforms. Since the initial report of RATS by Melfi et al. [1], various port placements have been proposed worldwide. To date, several “game-changers,” such as newly developed instruments and updates of robotic systems, have significantly influenced the alterations in the port locations. The prevalence of video-assisted thoracoscopic surgery (VATS) procedures, including uniportal techniques and improved surgeons’ skills, have been associated with the alterations.

In surgical view, the four-arm, look-up-view method, in which the intrathoracic cranial side is viewed from the caudal side, is considered the current mainstream approach worldwide. Many variations have recently emerged and evolved from this conventional method, including fewer ports or fewer skin incisions. In contrast, approaches in which the intrathoracic craniocaudal axis of the patient is aligned with the horizontal direction of the surgeon console monitor, such as the so-called horizontal open-thoracotomy-view approach or simply the horizontal-view approach, have recently been reported. We have classified the port variations into four categories considering the above points and the history of their advent. Furthermore, to visually and comprehensibly provide these variations for practical use, we herein present elaborate illustrations based on the descriptions on the reviewed reports, and discuss the characteristics of the representative approaches.

## 2. Material and Methods

### 2.1. Report Selection

This is a narrative review. All types of English literature (original articles, case reports, short reports on surgical techniques, etc.) describing the major port arrangements and approaches used for performing RLRs with the da Vinci Surgical System were considered. The inclusion criteria of the articles in this review are as follows: the article described RLRs and the description of the method, approach, and port placement, including figures, was detailed and clear. The articles that were based on a technique other than RLRs; provided unclear description of the method, approach, and port placement; and overlapped with the other representative reports that provided the sources of the methods were excluded from the study. In addition, articles that provided useful information to readers were included at the authors’ discretion.

We conducted a PubMed Advanced English literature search for selecting the papers published during a time frame from May 2002, i.e., when Melfi et al. first reported RLRs [1], to October 2022 using the following keyword combinations: “((RATS) OR (robot)) AND (lung) AND (approach),” 2698 results; “((RATS) OR (robot)) AND (lung) AND (resection),” 741 results; and “((RATS) OR (robot)) AND (lung) AND (port),” 166 results. As many articles as possible (2698 + 741 = 3439) were identified using the first two keyword combinations; we initially screened 166 reports using the automation search tool and last-term combinations.

Among the searched reports, 72 (nonrobotic, nonlung, or other) irrelevant reports were excluded and 94 reports were surveyed. Further, reports were extracted based on a review of their titles and abstracts; the full text of 44 articles was carefully reviewed, and articles that fully provided descriptions or figures of the port arrangements and approaches were selected. The authors thoroughly reviewed the selected reports, and their citing articles were surveyed. Hence, five citing reports were further read. Finally, 30 reports that described the port arrangements and approaches were selected for this review and included in the references, of which 21 were included in Table 1 [1,2,3,4,5,6,7,8,9,10,11,12,13,14,15,16,17,18,19,20,21]. A schematic of the literature search flow is shown in Figure 1.

### 2.2. Categorization

We categorized the port placement variations into four-phase groups from the viewpoint of the advent histories: (I) early era, three-arm technique with utility incisions; (II) four-arm, total port placement technique without robotic staplers; (III) four-arm technique using robotic staplers (current mainstream approach); (IV) maximizing the functional features of the Xi platform, including (IV-A) significant alterations in the view direction, such as the horizontal-view technique, and (IV-B) reducing the number of ports, including the ultimate single incision (uniport) technique.

### 2.3. Visualization

We created several elaborate illustrations based on the descriptions of the reviewed literature to comprehensively visualize the port placement and key points of each approach for practical use. Some parts or points, which were not described in the original reports, were not shown. For example, if the description of only one (right or left) side was recognized, the other unrecognized side was not described.

## 3. Results and Discussion

Figure 2, Figure 3, Figure 4, Figure 5, Figure 6, Figure 7, Figure 8, Figure 9, Figure 10, Figure 11, Figure 12, Figure 13, Figure 14, Figure 15, Figure 16, Figure 17, Figure 18 and Figure 19 and Table 1 were made based on descriptions from the included reports in this review. The figures are the most important components in this review. Table 1 complements the descriptive details.

### 3.1. Initial Phase: Three-Arm Procedure with a Utility Incision Derived from a Three-Port VATS Lobectomy Technique

The initial da Vinci system, described as “Standard” in Table 1, having three arms including two instrument arms and one camera arm was approved by the United States Food and Drug Administration in 2000. In this “early era” of robotic surgery, several pioneer thoracic surgeons developed RATS procedures with little or none of the published evidence. Port placement techniques during this early phase were mainly derived from the conventional three-port VATS lobectomy technique. A utility incision was shared by one of the robotic arms (instruments or camera) and a bedside assistant for retraction and stapling maneuvers.

Melfi et al. reported the first case series of RATS, including five lower lobectomy cases in 2002 (Figure 2) [1]. A 3cm utility incision (‘service entrance’) was created in the fourth intercostal space (ICS) for lung lobectomy. Subsequently, Ashton et al. reported their initial RATS lobectomy case using a five-incision and three-arm setting (Figure 3) [2]. Bodner et al. also reported one case of lower lobectomy and surgical treatment of nonpulmonary lesions using a similar setting [22]. All reported cases included patients who underwent lower lobectomies [1,2,22].

Park et al. reported the first consecutive case series of 34 patients with all types of lobectomies, including upper and middle lobes (Figure 4) [3]. In their procedures, the three incisions that were used for their VATS lobectomy were applied, because a possible conversion to the institution’s VATS procedure was considered without additional incisions. The main utility incision was placed at the level of the superior pulmonary vein in the mid-axillary line for upper lobectomy and one ICS below for middle or lower lobectomies. One of the limitations described by the authors was the importance of having assistants at the operating table who are familiar with conventional VATS lobectomy techniques, especially with regard to retraction and exposure of pertinent anatomic structures and stapling device introduction.

Gharagozloo et al. subsequently reported a similar case series of 61 patients, including all types of lobectomies (Figure 5) [4]. They utilized an additional 2cm port for retraction by bedside assistant, in addition to the three ports for robotic arms. They combined robotic surgeries with VATS. The authors used a robot to dissect the pulmonary artery and hilum. Once the dissection was completed, the robot was removed, and lobectomies were completed using VATS techniques.

### 3.2. Second Phase: Total Port (Four-Arm) Robotic Surgery without Using Robotic Staplers

The da Vinci S system was introduced in 2006. One of the major additional functions was the utility of the fourth robotic arm. The fourth arm was introduced by several surgeons to obtain a steady and controlled retraction of the lungs for better hilar dissection. In this phase, surgeons started the use of carbon dioxide (CO_2_) insufflation to maintain appropriate positive intrathoracic pressures.

Veronesi et al. first standardized a four-arm technique and reported their case series with the method (Figure 6) [5]. They placed the fourth port at the posterior seventh ICS using Cadiere forceps for retraction. The additional retraction by the fourth arm was emphasized to reduce the requirement to change instruments, avoid possible interferences between robotic and assistant instruments, and enable the direct control of retraction by a console surgeon.

Ninan and Dylewski first reported their technique of RATS without a utility incision [6], which was later updated by Dylewski et al. [7] (Figure 7) with a large cohort of patients who underwent anatomical lung resection by RATS (n = 200). They developed this technique with the subcostal assistant port that can access the thoracic cavity through the attachment of the diaphragm to the tenth rib. The CO_2_ insufflation effectively mobilized the diaphragm to the caudal side, enabling the use of an assistant port close to the diaphragm. Additionally, this technique placed all three or four robotic ports, including the camera and two or three instrumental ports, in the same ICS along the major pulmonary fissure, which may reduce injury to multiple intercostal neurovascular bundles.

Cerfolio et al. developed techniques in RATS lobectomy without a utility incision, which was named a completely portal robot lobectomy with four arms (CPRL-4) (Figure 8) [8]. They reported their large case series (n = 106) and comparison with propensity-matched patients who underwent lobectomy via rib- and nerve-sparing thoracotomy. CPRL-4 was associated with reduced morbidity, lower mortality, improved mental quality of life, and shorter hospital stay. Technical modifications, including the addition of a robotic retraction arm, vessel loop to guide the stapler, tumor removal above the diaphragm, and CO_2_ insufflation, reduced the operative time and conversion to thoracotomy. Cerfolio et al. also reported their methods in detail in other reports [23,24,25]. Detailed reviews of these representative settings and procedures were also reported [26,27].

### 3.3. Third Phase: Total Port (Four-Arm) Robotic Surgery Using Robotic Staplers

Avent of da Vinci X and Xi systems led to substantial advancements in RATS. The utility of robotic staplers was one of the major changes in X/Xi from S/Si platforms. The use of robotic staplers by the console surgeon has two significant advantages compared to the stapling maneuvers by bedside assistants: 1) the operating surgeon can fully control dividing procedures of the hilar vessels, bronchus, and lung parenchyma, which are crucial steps in anatomical lung resection; and 2) the articulation of robotic staplers is omnidirectional and better than endoscopic staplers which only bend single-direction. Having a sufficient distance between the pulmonary hilum and the port for the robotic stapler would commonly be better because the end portion of the robotic stapler, which consists of a cartridge and an anvil of the stapler, is much longer than other robotic instruments.

Pearlstein demonstrated their techniques for robotic lobectomy using robotic staplers, with the illustration of their port placement separately for upper lobectomy and middle/lower lobectomies (Figure 9) [9]. They recommended locating the port for staplers as low as the caudal as possible (close to the diaphragm) to allow the greatest maneuverability in the thoracic cavity.

Kim et al. (Figure 10, R) [10] and Khan et al. (Figure 10, L) [11] developed a “five on a dice” port placement, which was described to allow for full control of operative procedures by the console surgeon. Both groups emphasized the ability to fully control stapling by the console surgeon, which could allow surgeons to have limited assistance during RATS.

### 3.4. Fourth Phase: Maximizing the Functional Features of the Xi System

The updated da Vinci robotic system, Xi, has several advancements from S/Si in addition to the introduction of the robotic stapler. One of these advancements is the improved latitude of robotic arms based on reduced external arm collision and flexibility of port placements brought by thinner arms, longer instruments, and the accessibility from multiple directions. The versatility of the Xi system allowed surgeons to develop multimodal and unique technical advancements, including challenges (A) to adopt the so-called horizontal open-thoracotomy-view technique in RLR, and more recently, (B) to reduce the number of ports, including uniport RATS, as described below.

#### 3.4.1. Fourth Phase A: Horizontal Open-Thoracotomy-View Approaches

This type of approach was mainly reported in Japan, and can provide views in which the intrathoracic craniocaudal axis of the patient is aligned with the horizontal direction of the surgeon console monitor screen, which is named the horizontal open-thoracotomy-view approach or simply the horizontal-view approach. As VATS has several approaches, including the look-up-view method and the confronting monitor method, it is natural that some surgeons prefer to perform the procedure with the horizontal-view similar to their familiar thoracotomy surgery, even in RLRs. This type of approach can be explained as a modified conversion of the robotic mediastinal surgery in the supine position to lung resection in the lateral decubitus position. Cranially located intrathoracic structures or instrument tips, which are sometimes hidden and difficult to confirm in the look-up-view method, were visually confirmed in the front. On the other hand, the robotic scope and the target structures or lungs are often in close proximity, occasionally resulting in the limited maneuverability of the scope and the instruments in comparison with conventional look-up-view methods. Especially in the four-arm setting, the nonoperating retraction arm can easily be obscured from view, although quantitatively demonstrating this characteristic is difficult. Specific technical considerations are needed because surgical views and settings are different from those in the well-established worldwide conventional approach.

Yamazaki et al. reported the “anterior approach” to RLR using the da Vinci Si System. This approach always provides a horizontal view of the intrathorax from the ventral side, regardless of the side to be operated on (Figure 11) [12,13]. They compared this technique with the conventional look-up method at their institution and reported shorter operative time, less blood loss, and fewer postoperative complications in the former technique than in the latter, especially for left upper lobectomy cases.

Funai et al. reported on a four-arm, horizontal-thoracotomy-view method using the Xi system for the right upper lobectomy case, and they called this the “Hamamatsu method” after their institution’s name (Figure 12) [14]. They reported a change of port distribution from the conventional Cerfolio’s CPRL-4 technique on the right side surgery, and the methodology for the left side surgery was not provided.

Sakakura et al. reported “the three-arm, robotic open-thoracotomy-view approach” using vertical port placement and a confronting upside-down monitor setting to perform RLRs (Figure 13) [15,16]. The direction of the view is always from the right side to the left side of the patient, regardless of the side to be operated on. Their method differs from the four-arm method of Yamazaki and Funai in that it utilizes three-arm and confronting monitor settings, and it always requires two assistants. Although they continued using their three-arm and confronting monitor settings, they mentioned that their three-arm technique may be meaningful as an introduction to the four-arm setting. The authors reported their method focusing on segmentectomy procedures [16], and also described it focusing on possible emergency rollout procedures [28]. The left intrathoracic view of Sakakura’s approach is similar to that of Yamazaki’s, and the right intrathoracic view of Funai’s and Sakakura’s is opposite to that of Yamazaki’s. These approaches may be more advantageous for upper lobe surgeries than for lower lobe procedures considering the recent increase in segmentectomies [16]. Conversely, the conventional look-up-view procedure can be more advantageous for complex single-direction segmentectomies of the basal segments [29,30]. Pain may be enhanced considering that the ports of these types of procedures were placed across multiple intercostal spaces. The assist port and its location may also affect pain. Hence, quantitative investigations for pain would be required for future studies.

**Figure 19 jpm-13-00230-f019:**
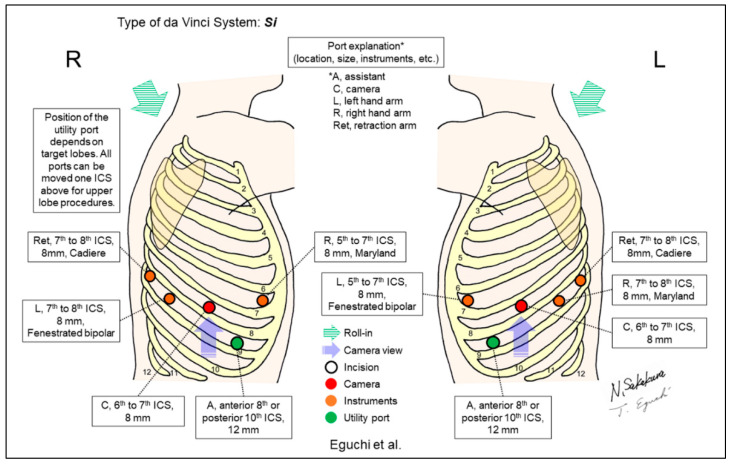
A modification of Cerfolio’s setting (T.E., Shinshu University, unpublished) [29,30].

#### 3.4.2. Fourth Phase B: Reducing the Number of Ports

Kang reported their anterolateral approach for removing painful posterior ports and decreasing the total number of ports (Figure 14) [17]. The effective use of the fourth robotic arm and utility port shared by one robotic instrument and the bedside assistant minimized the assistants’ role, and enabled a solo surgery platform.

Ueda et al. developed a three-incision RLR with the development of one incision that is shared by a robotic camera and one robotic instrument, and the reduction of the fourth port with the use of a well-collaborated assistant utility port manipulated by a bedside assistant, who can freely assist surgery without interference from the robotic arms based on port locations (Figure 15) [18]. A 4cm incision was shared by two robotic arms (scope and instrument). Interference between the two arms was prevented using the “para-axial method.”

Han et al. developed a two- or three-port approach using three robotic arms (Figure 16) [19]. Patients who underwent their two-port approach were compared with their previous three-port system, resulting in significantly less postoperative pain after the two-port approach than after the three-port approach. A multi-instrument laparoscopic port was used in their two-port approach to share a 3–4cm incision with a robotic scope, a robotic instrument, and an additional assistant instrument.

Recent advances in the uniportal VATS technique led some surgeons to develop the uniportal RATS. Categorizing the camera direction may be difficult in the uniportal setting as either the look-up view or the horizontal view. Yang et al. first reported a case who underwent a right upper lobectomy with the uniportal RATS technique, in which a single 4–4.5cm incision was created to be shared by three robotic arms and the assistant (Figure 17) [20]. They used 8mm trocars for all robotic arms to fit the limited incision. This setting restricted the use of robotic staplers.

Most recently, Gonzalez-Rivas et al. reported their uniportal RATS techniques using robotic staplers (Figure 18) [21]. They described that the technique has been applied not only to lobectomies but also to complex lung resections, including segmentectomy, pneumonectomy, and bronchoplasty. Based on their skills and experience of using uniportal VATS, the authors described that an advantage of uniportal RATS has the availability of quick emergent conversion to the uniportal VATS or anterior thoracotomy. In the future, da Vinci Single-Port “SP,” which is currently being trialed in the thoracic surgery field, is expected to be put to practical use for RLRs.

We herein reviewed representative reported port variations for RLRs and provided several elaborate illustrations for practical and comprehensible use. Other variations and modifications based on previously reported methods must be present, as well as some excellent unique approaches at individual institutions although they have not yet been reported regardless of various approaches, such as the conventional look-up-view types, the fewer-arm or fewer skin incision types derived from the conventional procedures, the horizontal-view types, and the newer reduced port or uniport types. For example, T.E. prefers a modified Cerfolio method in his institution (unpublished data) (Figure 19). This method involves moving the three ventral ports (camera, anterior instrumental, and utility ports) from the Cerfolio setting to the one or two ICS cranial side to avoid port collisions, considering the smaller body sizes of Asians compared to Westerners.

As a limitation, these approaches presented herein have specific characteristics, advantages, and limitations. Although these procedures have emerged after several improvements and innovations made by the expert surgeons, it should be noted that some of these procedures considerably differ from the conventional procedures recommended by the product company (e.g., maintaining a distance between the target and the camera, providing a certain space between the ports, etc.). The uniportal setting significantly deviates from the recommended setups.

In conclusion, thoracic surgeons should be thoroughly versed in these approaches so they can choose the approach that best suits the case and their preferences. We hope that this review will be of help in this respect.

## Figures and Tables

**Figure 1 jpm-13-00230-f001:**
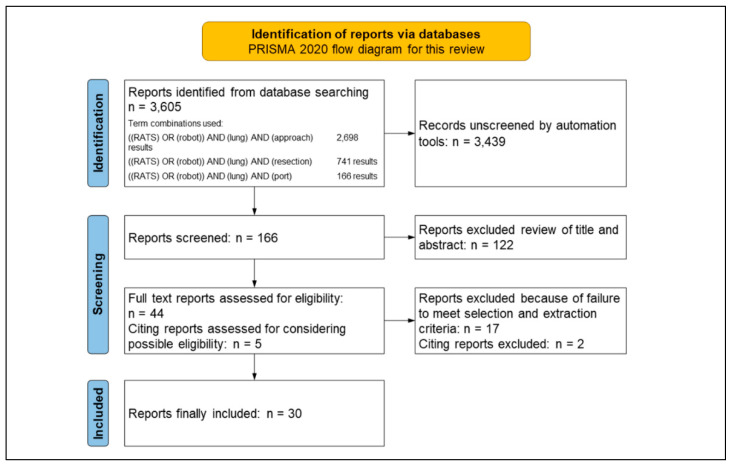
Flow diagram for selecting reports.

**Figure 2 jpm-13-00230-f002:**
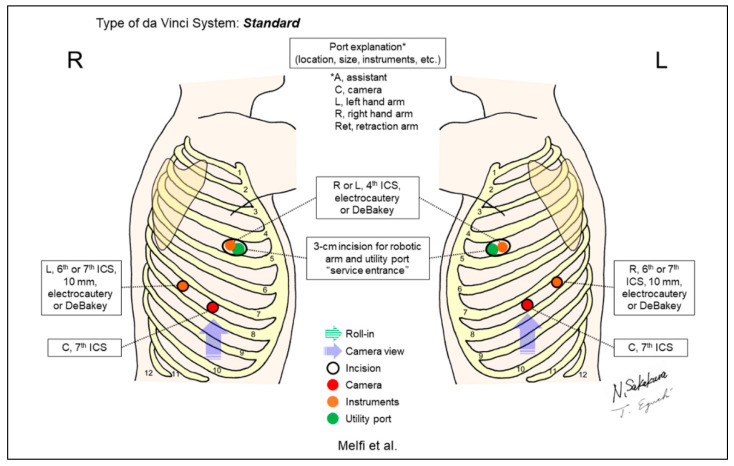
Melfi et al.’s setting [1].

**Figure 3 jpm-13-00230-f003:**
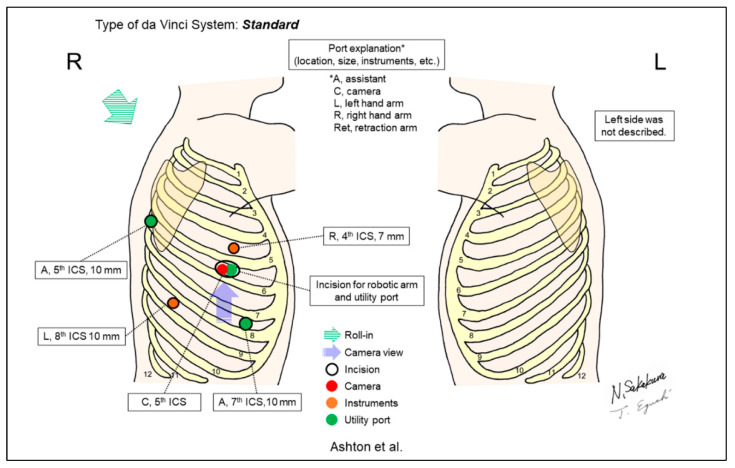
Ashton et al.’s setting [2].

**Figure 4 jpm-13-00230-f004:**
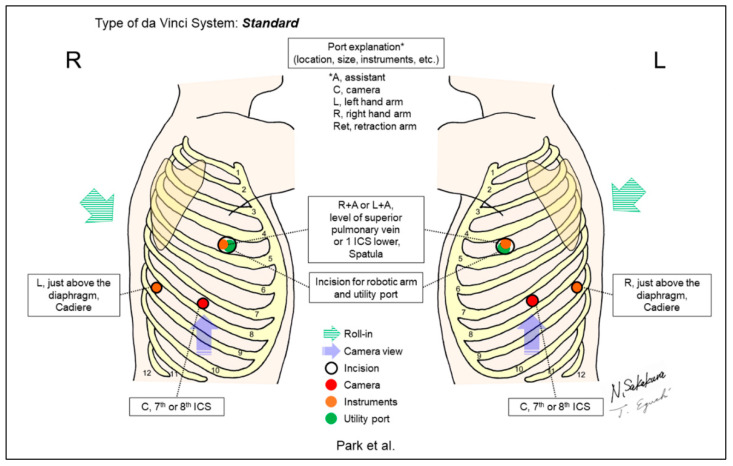
Park et al.’s setting [3].

**Figure 5 jpm-13-00230-f005:**
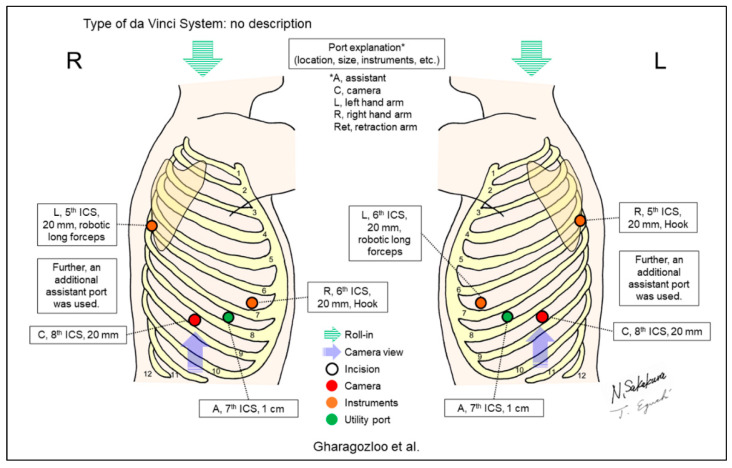
Gharagozloo et al.’s setting [4].

**Figure 6 jpm-13-00230-f006:**
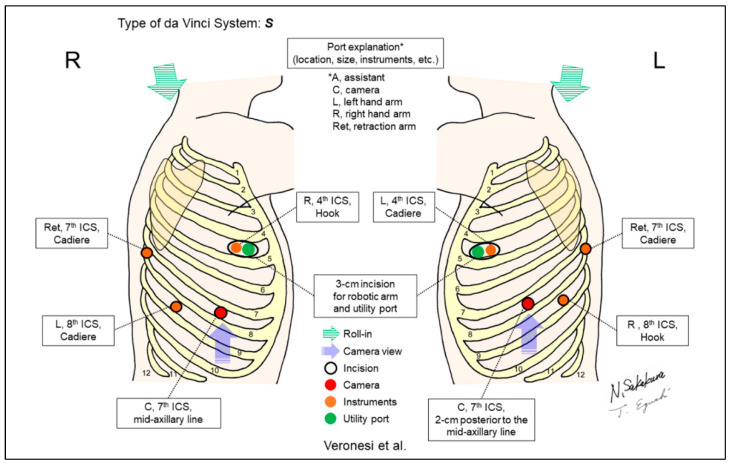
Veronesi et al.’s setting [5].

**Figure 7 jpm-13-00230-f007:**
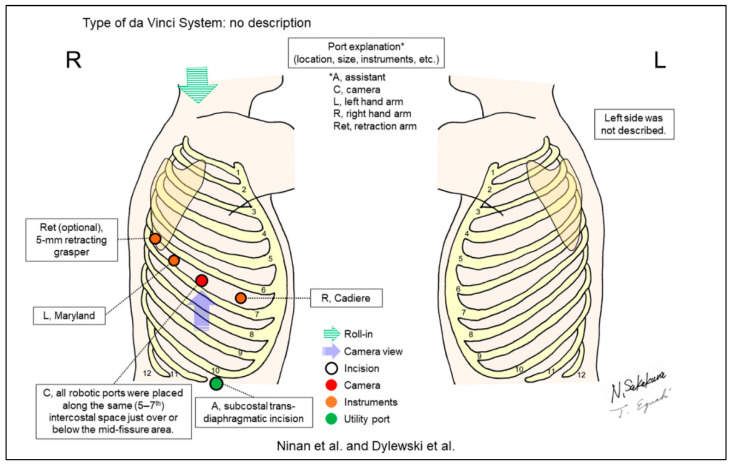
Ninan et al.’s [6] and Dylewski et al.’s [7] setting.

**Figure 8 jpm-13-00230-f008:**
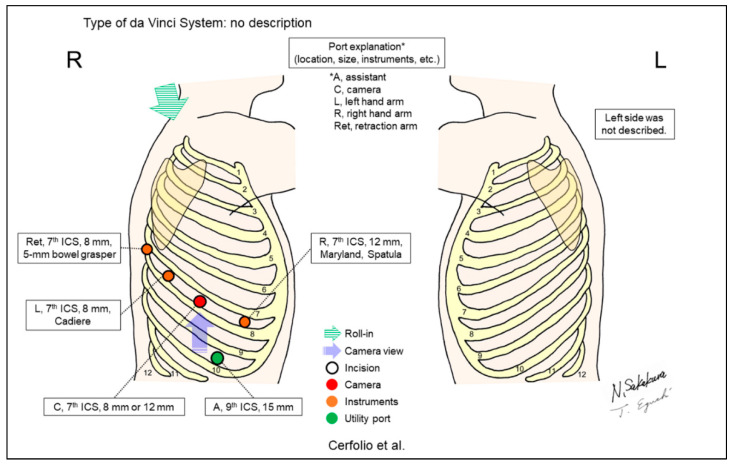
Cerfolio et al.’s setting [8].

**Figure 9 jpm-13-00230-f009:**
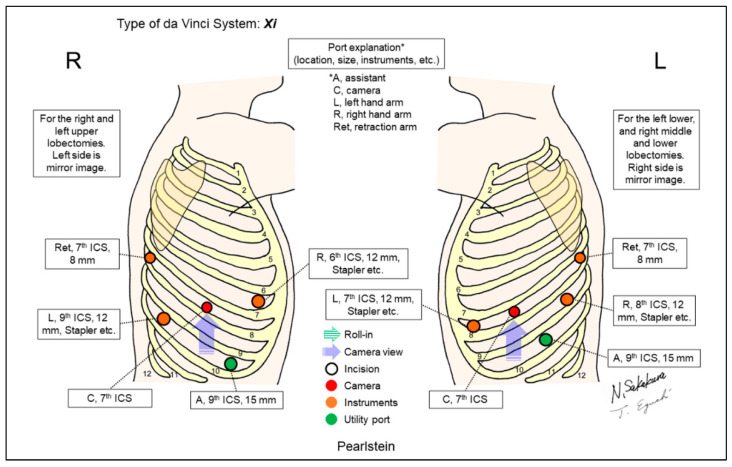
Pearlstein’s setting [9].

**Figure 10 jpm-13-00230-f010:**
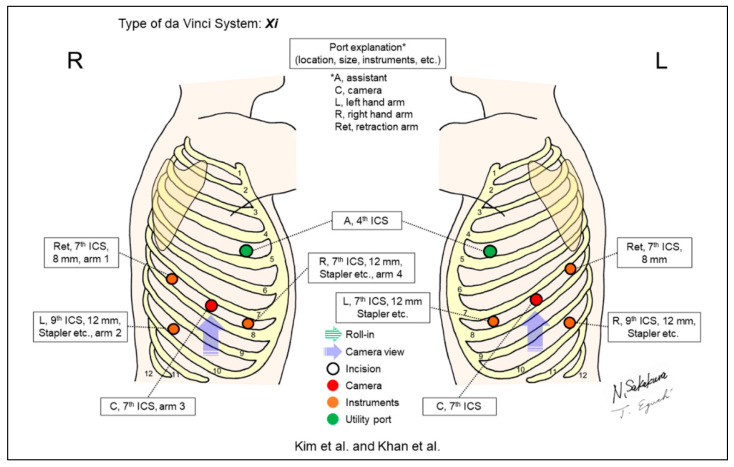
Kim et al.’s [10] (R) and Khan et al.’s [11] (L) setting.

**Figure 11 jpm-13-00230-f011:**
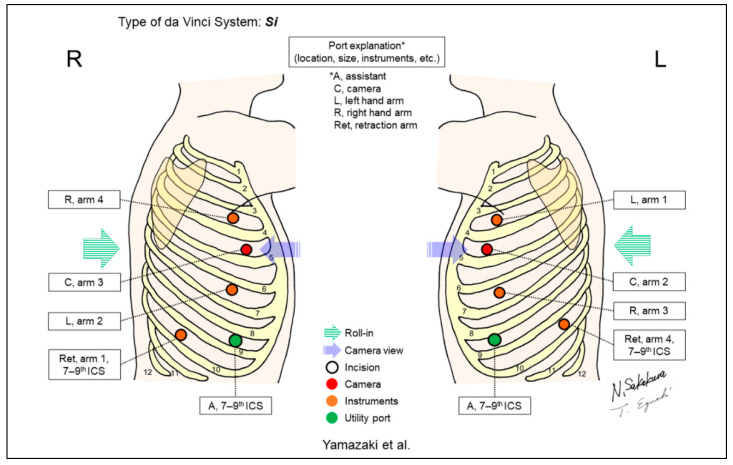
Yamazaki et al.’s setting [12,13].

**Figure 12 jpm-13-00230-f012:**
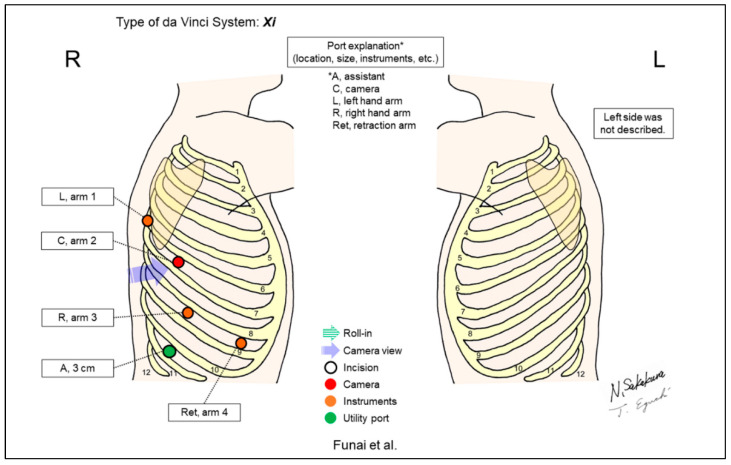
Funai et al.’s setting [14].

**Figure 13 jpm-13-00230-f013:**
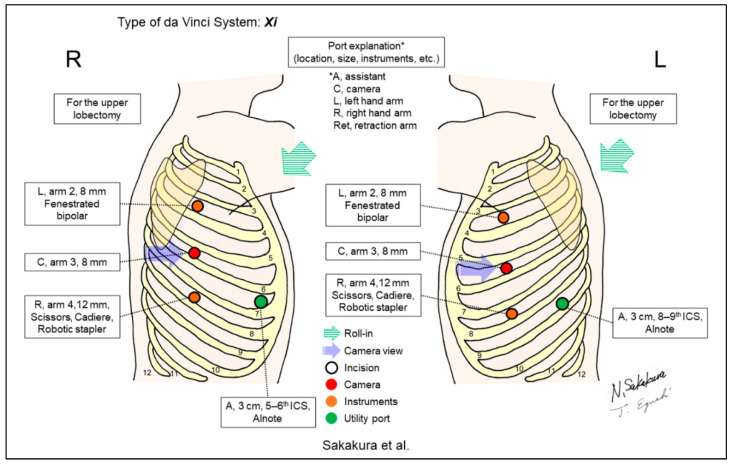
Sakakura et al.’s setting [15,16].

**Figure 14 jpm-13-00230-f014:**
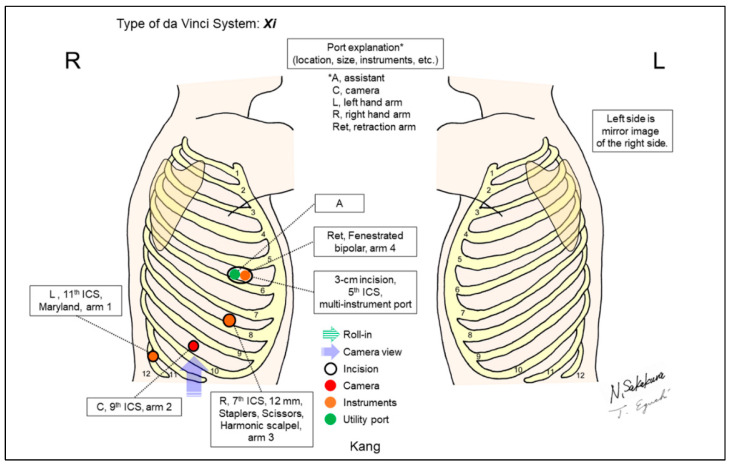
Kang’s setting [17].

**Figure 15 jpm-13-00230-f015:**
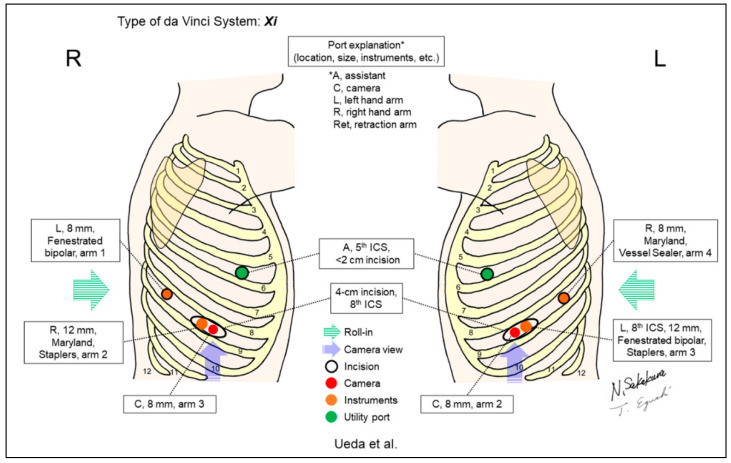
Ueda et al.’s setting [18].

**Figure 16 jpm-13-00230-f016:**
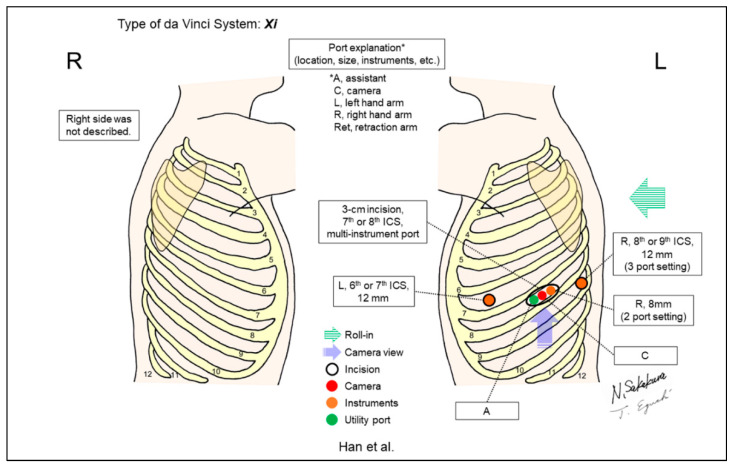
Han et al.’s setting [19].

**Figure 17 jpm-13-00230-f017:**
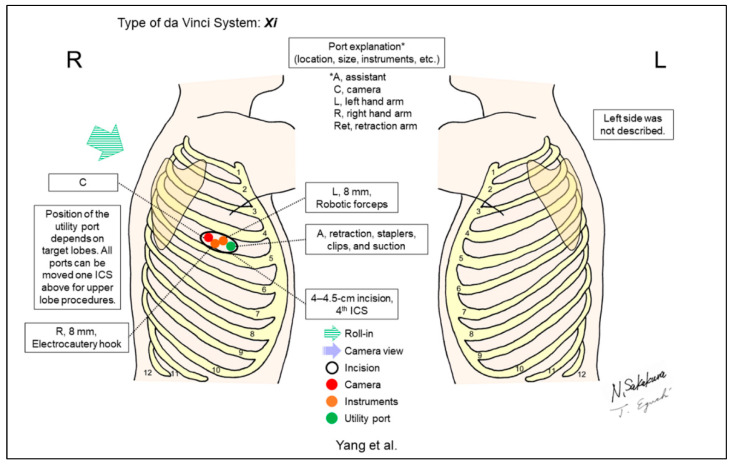
Yang et al.’s setting [20].

**Figure 18 jpm-13-00230-f018:**
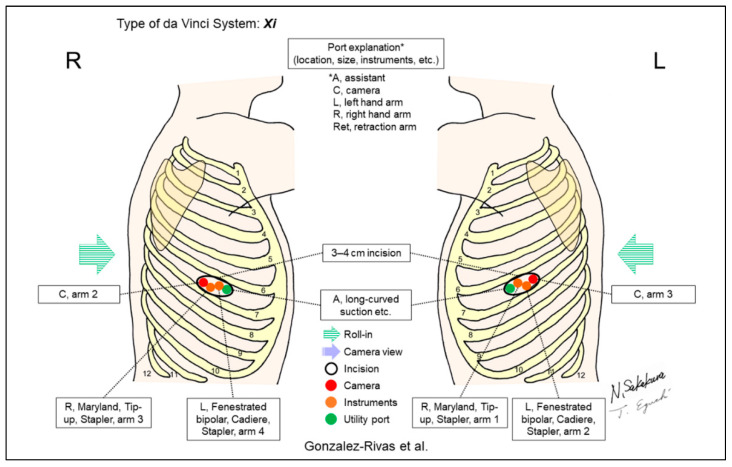
Gonzalez-Rivas et al.’s setting [21].

**Table 1 jpm-13-00230-t001:** Reports for robotic port placement for anatomical lung resection.

First Author/Year [Reference]	da Vinci System	View Type	Number of Skin Incisions ^a^	Number of Robot Arms ^b^	Scope Type	Location of Scope (ICS)	Location of Utility Port for Assistant (ICS)	CO_2_ Insufflation	Number of Patients [Reference]	Features
Melfi 2002 [1]	Standard ^c^	Look-up	3	3	0-degree	7th	4th	No	12 (5 lower lobectomies)	The first report on robotic surgery for thoracic diseases. A utility incision “service entrance” is placed at the 4th ICS, which is shared for an assistant and a robotic arm.
Ashton/2003 [2]	Standard ^c^	Look-up	5	3	0-degree	5th	3 utility ports, anterior 5th, 7th, and posterior 5th	No	1 (lower lobectomy)	Three utility incisions were placed at the anterior 5th, 7th, and posterior 5th ICSs. The anterior 5th ICS was shared with a robotic scope.
Park/2006 [3]	Standard ^c^	Look-up	3	3	30-degree, down/up	7–8th	Superior pulmonary vein level or one ICS lower	No	34	The incisions used for authors’ thoracoscopic lobectomy were applied. A utility incision is placed at the level of the superior pulmonary vein or one ICS lower, which is shared for an assistant and a robotic arm.
Gharagozloo/2008 [4]	ND ^d^	Look-up	4	3	ND	8th	ND	ND	61	A hybrid robotic-VATS technique. In addition to 3 robotic ports, a 1cm assistant port for an endoretractor was placed. Authors used a robot to dissect the pulmonary artery and the hilum. Once dissection was completed, the robot was withdrawn, and lobectomy was completed by VATS.
Veronesi/2010 [5]	S ^e^	Look-up	4	4	ND	7th	4th	No	54	A posterior port for the 4th retraction arm was placed. The additional retraction by the 4th arm could reduce the requirement to change instruments and avoid possible interferences between robotic and assistant’s instruments.
Ninan/2010 [6] Dylewski/2011 [7]	ND ^d^	Look-up	4 or 5	3 or 4	0-degree	5–7th	Subcostal	Yes	74 [6], 200 [7]	The “total endoscopic robotic video-assisted approach.” The subcostal trans-diaphragmatic incision was placed for a utility port by a bedside assistant.
Cerfolio/2011 [8]	ND ^d^	Look-up	5	4	ND	7th	9–10th	Yes	106 (robotic lobectomies)	The “completely portal robotic pulmonary lobectomy with 4 arms (CPRL-4).”
Pearlstein/2016 [9]	Xi	Look-up	5	4	0-degree	7th	9th	ND	ND	Specific techniques to use robotic staplers were described. Locating the stapling ports as low caudal as possible (close to the diaphragm) allows the greatest degree of maneuverability of the stapler in the chest.
Kim/2017 [10] Khan/2017 [11]	Xi	Look-up	5	4	30-degree ^c^	7th	4th	Yes	ND	“Five on a dice” method. Improved control of vascular stapler via inferior ports in the both sides and good retraction of the lung using tip-up grasper with sponge. A challenging robotic left pneumonectomy was performed [11].
Yamazaki/2020 [12], 2021 [13]	Si	Horizontal	5	4	30-degree, down	4th	7–9th	Yes	87 (anterior approach) [13]	“Anterior approach” technique. Intrathorax is always viewed from the ventral side of the patient regardless of the side to be operated on.
Funai/2020 [14]	Xi	Horizontal	5	4	ND	5th	10th	Yes	ND	A four-arm, horizontal-view approach, the “Hamamatsu method.” A change of port distribution from the conventional Cerfolio’s CPRL-4 technique was shown on the right-side surgery.
Sakakura/2021 [15], 2022 [16]	Xi	Horizontal	4	3	30-degree, down/up	Right 5–6th, Left 4–6th	Right 5–7th, Left 7–9th	Yes	58 [15], 114 [16]	“Three-arm, open-thoracotomy-view approach” using vertical port placement and confronting up-side down monitor setting. All surgeons obtain “bird-eye” views as though they perform thoracotomy surgery. Ventral/dorsal hilum becomes visible by switching the 30-degree camera down/up.
Kang/2019 [17]	Xi	Look-up	4	4	ND	9th	5th	Yes	36	“Anterolateral approach.” A utility port in the 5th ICS shares with a robotic arm. Fully use the 4-arm technique, minimize the assistant’s role and establish a solo surgical method, and avoid painful posterior ports.
Ueda/2021 [18]	Xi	Look-up	3	3	30-degree, down	8th	5th	No	39	A “three-incision robotic surgery.” A 4cm incision was shared for two robotic arms (scope and instrument). Interference between the two arms were prevented by “para-axial method.”
Han/2022 [19]	Xi	Look-up	2 or 3	3	ND	7–8th	7–8th	Yes	142	A matched analysis for “two-port” and “three-port” approaches. In the two-port setting, a 3–4cm working port was shared for two robotic arms (scope and instrument) and an additional assistant instrument using multi-instrument laparoscopic port.
Yang/2021 [20]	Xi	Undefined ^f^	1	3	30-degree	4th	4th	No	1 (right upper lobectomy)	Uniportal RATS. A single 4–4.5cm incision was created to be shared by three robotic arms and the assistant. Robotic arms were intercrossed inside the chest and the control of the arms needed to be reset on the console accordingly.
Gonzalez-Rivas/2022 [21]	Xi	Undefined ^f^	1	3	ND	ND	ND	No	ND	Pure uniportal RATS. To avoid collision, cancel arm 1 on the right side (arm 2 for camera) and arm 4 on the left side (arm 3 for camera). The camera is placed in the posterior part of the incision to allow the other two robotic instruments to work. All types of lung resections, including segmentectomies, sleeves and carinal resections were performed.
Eguchi (Shinshu University, unpublished data)	Si	Look-up	5	4	0-degree	6–7th	7–10th	Yes	180 (50 lobectomies, 130 segmentectomies, unpublished data)	A modification of Cerfolio’s setting. Moving the three ventral ports from the Cerfolio setting to the one or two ICS cranial side to avoid port collisions, considering the smaller body sizes of Asians compared to Westerners.

^a^ All skin incisions including a utility port. ^b^ Operating arms including robotic scope and instruments. ^c^ Initial da Vinci Surgical System. ^d^ The original report simply described the “da Vinci Surgical System” as being used [5,7,8,9]. ^e^ Not described in text but recognized on a picture or a movie adopted in the original report. ^f^. In uniportal settings, categorizing the camera direction may be difficult as either the look-up view or the horizontal view. ICS, intercostal spaces; ND, no description; RATS, robot assisted thoracoscopic surgery, VATS, video assisted thoracoscopic surgery.

## Data Availability

Data sharing is not applicable.

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
