# Peer review of "Port Placement Variations for Robotic Lung Resection: Focusing on Their History, Conventional Look-Up-View and Horizontal Open-Thoracotomy-View Techniques, and More"

_jpm, 2023, doi:10.3390/jpm13020230_

Round 1

Reviewer 1 Report

Dear Authors,

thanks for submitting this interesting manuscript focused on the evolution of port placement in robotic thoracic surgery. 
I would suggest to improve the quality of your manuscript improving the methodology section and stating if is it a narrative or systematic review, the time frame of the selected papers, the inclusion and exclusion criteria of your search.
Thanks again

Author Response

We sincerely appreciate and thank you for highly valuable comments and constructive suggestions. We have tried to revise the manuscript in order to make the necessary corrections in line with the reviewers’ suggestions and comments as much as possible. All corrections in the revised manuscript are indicated in yellow highlight.

Comment 1: Reviewer 1 suggested that the methodology section be revised to indicate the type of review, time frame, and criteria for inclusion or exclusion of references.

(Original comments: “Dear Authors, thanks for submitting this interesting manuscript focused on the evolution of port placement in robotic thoracic surgery. I would suggest to improve the quality of your manuscript improving the methodology section and stating if is it a narrative or systematic review, the time frame of the selected papers, the inclusion and exclusion criteria of your search. Thanks again.”)

Answer 1: Thank you, we agree with the reviewer’s viewpoints. We have revised the methodology section and have clarified the type of this review, time frame, and criteria for inclusion or exclusion of the references.

Change 1: lines 2-3; lines 24-26; lines 46-61

We have also added five additional references as required by the journal and for readers’ understanding [10-14].

Thank you again for your highly valuable comments and constructive suggestions. We hope that the manuscript is satisfactory. We would like to express our sincerest gratitude for your kind consideration of our manuscript.

Reviewer 2 Report

Thank you for the opportunity to revise such an interesting paper.

Congratulations on your effort to recreate the history and trends in this subject.

I only suggest a final revision due to minor spelling mistakes.

BW

Author Response

Response to Reviewer 2

We sincerely appreciate and thank you for highly valuable comments and constructive suggestions. We have tried to revise the manuscript in order to make the necessary corrections in line with the reviewers’ suggestions and comments as much as possible. All corrections in the revised manuscript are indicated in yellow highlight.

Comment 1: Original comments: “Thank you for the opportunity to revise such an interesting paper. Congratulations on your effort to recreate the history and trends in this subject. I only suggest a final revision due to minor spelling mistakes. BW”

Answer 1: The final version of the manuscript was thoroughly reviewed by the native English editors.

We have also added five additional references as required by the journal and for readers’ understanding [10-14].

Thank you again for your highly valuable comments and many constructive suggestions. We hope that the manuscript is satisfactory. We would like to express our sincerest gratitude for your kind consideration of our manuscript.

Reviewer 3 Report

Dear authors,

thank you for submitting the paper. Very good overview about the possible port placement variations.

Can you please critital discuss under the aspect of liablity and legal basis the authorized and/ or recommended port placements by the company. This needs at least be mentionend.

Author Response

Response to Reviewer 3

 We sincerely appreciate and thank you for highly valuable comments and constructive suggestions. We have tried to revise the manuscript in order to make the necessary corrections in line with the reviewers’ suggestions and comments as much as possible. All corrections in the revised manuscript are indicated in yellow highlight.

Comment 1: Reviewer 3 suggested that the basic procedures recommended by the product company need to be mentioned.

(Original comments: “Dear authors, thank you for submitting the paper. Very good overview about the possible port placement variations. Can you please critical discuss under the aspect of liability and legal basis the authorized and/or recommended port placements by the company. This needs at least be mentioned.”

Answer 1: Thank you, we agree with the reviewer’s comments. We have added this point in the revised manuscript in the last of comment section.

Change 1: Lines 276-282

We have also added five additional references as required by the journal and for readers’ understanding [10-14].

Thank you again for your highly valuable comments and many constructive suggestions. We hope that the manuscript is satisfactory. We would like to express our sincerest gratitude for your kind consideration of our manuscript.